# Choreographing rhizosphere effect with agricultural practices for agroecology?

**Edith Le Cadre**[1]*, **Sebastian Mira**[1,2], **Xiaoyan Tang**[3], **Mathieu Emily**[4]

**1** Institut Agro, INRAE, SAS, Rennes, France, **2** Eurede, Eureden Innovation, Landerneau, France, **3** College of Resource, Sichuan Agricultural University, Chengdu, China, **4** Institut Agro, Univ Rennes1, CNRS, IRMAR, Rennes, France

* edith.lecadre@institut-agro.fr

## Abstract

For sustainable agriculture, soil-plant interactions (i.e., the rhizosphere effect) is prominent focus, since they determine plant health and nutrition. However, system-level agricultural management practices interfere with the rhizosphere effect. In this study, we characterized the rhizosphere effect of winter wheat (*Triticum aestivum* var. LG Absalon) on farm fields along two levels of crop diversity induced by rotation (high or low) combined with two tillage intensities (conventional or reduced). The rhizosphere effect was determined from differences in enzyme activities involved in carbon (β-glucosidase), nitrogen (arylamidase) and phosphorus (acid phosphatase) cycles measured in the rhizosphere and bulk soil. We observed positive rhizosphere effects for all enzymes, but they were significantly altered by soil tillage. High temporal diversification and reduced tillage increased the intensity of the rhizosphere effect for all enzymes studied, suggesting the relevance of agroecological management of arable land to promote nutrient cycling. In contrast, benefits of crop diversification on the rhizosphere effect decreased drastically under conventional tillage. Accordingly, the rhizosphere effect should be carefully synchronize with agricultural practices under agroecological transition.

## 1. Introduction

Nature-based solution can achieve a triple benefit (i) resilience and mitigation of climate change, (ii) productivity and (iii) preservation of biodiversity and other natural resources [1]. In farmlands nature-based solution or agroecology is endorsed as the only pathway for tackle future human needs [2]. Agroecological solutions are fueled by plant-soil feedback which determine productivity and multifunctionality [3,4]. Accordingly, agronomists have paid more attention to plant-soil feedback [5–9] to better plan and manage biodiversity in fields and over time [10], such as using relay cropping, intercropping, agroforestry or perennial cropping to address sustainability issues [11].

**Data availability statement:** The dataset is available at: Le Cadre-Barthélémy E (2025) Soil enzymes. V1 edn. Recherche Data Gouv. doi:10.57745/NFQNIY Repository : Recherche Data Gouv.

**Funding:** This project was financially supported by the Chaire Agriculture Ecologiquement Intensive, subprogram Rhizosphere. Sebastian Mira was funded for his salary by EUREDEN with CIFRE N° grant 2019/ 0828. Funders played no role in the study design, data collection and analysis, decision to publish or preparation of the manuscript.

**Competing interests:** The authors have declared that no competing interests exist.

Plant-soil feedback occurs mainly along two pathways: the litter (indirect) and the rhizosphere (direct) [12]. Although the rhizosphere is limited to the small soil volume around living roots, there is mounting evidence that the rhizosphere effect is of major importance [13–15] to control legacy, magnitude, direction of plant-soil feedbacks. The rhizosphere effect increases the availability of nutrients [16] or plant health [17–20]. Accordingly, managing the rhizosphere effect of plant-soil feedback under field conditions is crucial for farmers to adopt nature-based solutions [21]. To enhance the rhizosphere effect, focal plants have to be reconnected during their life-times to soil microbial communities that depends greatly on the patterns of rhizode-position [22–24] that feed root-associated microbes [25]. Rhizodeposition is a part of plant strategies to acquire resources [26–28] that triggers microbial diversity [29,30], extracellular enzyme activity [31,32] and soil nutrient availability and changes in pH [31,33]. Beyond the plant-species level, interspecific factors such as plant diversity and the associated variability in root functional traits can also influence rhizodeposi-tion patterns [27,34]. Plant diversity in farmlands could be increased by addition of more either by temporal patterns, such as crop rotation, and spatial configurations, such as intercropping, relay cropping, and strip cropping. These patterns of crop diversification have been identified that enhance microbial rhizosphere diversity and carbon (C) inputs [35–38]. However, it remains unclear whether plant functional identity (e.g., fast vs. slow growth), performances in mixtures (e.g., trait plasticity) or complementarity among individuals (e.g., trait complementary) have more influence on the rhizosphere effect than site-specific conditions (i.e., soil abiotic and biotic properties) [3,37,39–47].

Inherent soil properties created by pedogenetic processes also determine the range of the rhizosphere effect [48] by influencing geochemistry [49] and the abun-dance and activity of soil microbial communities [50]. Other variables derived from soil properties, such as soil structure, temperature and water holding capacity, are intrinsically related to intensity of the rhizosphere processes by controlling nutrient diffusion, mass flows and mineralization of organic matter [51]. Crop diversification patterns (rotational or spatial configurations) and other agricultural practices (fertil-ization, irrigation, pesticide, tillage) can alter any of these soil properties [52–54] and thus in turn the reservoir of microorganisms initially present in soils from which plants select a specific microbiome of the rhizosphere [55]. Managing the rhizosphere effect requires therefore an holistic system-level approach to illuminate the ability of culti-vated plants to adapt to their environment.

Rhizosphere enzymes are considered as a footprint of plant-microbe interactions [16,55–59]. Due to their ubiquity and responsiveness to land-use changes, soil enzymes have been used as indicators to evaluate impacts of agricultural practices [37,60] on biogeochemical cycles [61], ecosystem stability in the face of extreme weather events [62] and C stocks [27]. Based on these studies, determining bulk soil and rhizosphere enzyme activities can reveal combined effects of plant diversity and management practices [57,58,59,62–66]. In addition, soil enzymes can provide the simple, inexpensive and repeatable measurements [67] that farmers require to obtain information about their practices in order to transition towards nature-based solutions.

In this study, we hypothesized that plant diversification by rotation and reduced tillage by preserving soil habitat for microbes would enhance the rhizosphere effect. As a proof of concept, we chose to locate our study in a specialized territory shaped by livestock and intensive practices, the northwestern France (Brittany), to determine if rhizosphere management is possible. Three enzymes (β-glucosidase, arylamidase and acid phosphatase) were measured in each rhizosphere and standardized (by water content and temperature) bulk soil compartments to estimate a relative rhizosphere effect. We then analyzed the influence of (i) crop diversification, and (ii) soil disturbance (here, tillage) on the rhizosphere effect on six farms. We finally collected the farmer's perception by focus group and analyzed their willing to modify their practice based on rhizosphere effect determination.

## 2. Materials and methods

### 2.1. Study sites

The experiment was conducted during spring and summer in 2021 in Northwestern Brittany characterized a by temperate oceanic climate. Six study sites (S1-S6, corresponding to one field of 6 different farms), with similar soil properties, were all cultivated in wheat (Triticum aestivum, var LG ABSALON). Sites were selected for their similar soil properties and pre-crop (maize, *Zea mays*) to limit variability when comparing rhizosphere among sites (Table 1). All farms were managed as typical conventional cropping systems in Brittany: maize (*Zea mays*) was the pre-crop, which was followed by the same elite wheat (*Triticum aestivum*) cultivar (LG Absalon). This cultivar was chosen in consensus with all farmers, in accordance with their objectives and criteria for precocity and pest resistance. No organic fertilizers were applied during wheat development, but synthetic fertilizers and fungicides were (Table 1), but no less than six weeks before soil sampling. All farmers willingly accepted the access to their fields.

To capture the legacy effect of past crops modifying soil properties including the soil microbiome [68–71], we calculate the Crop Diversity Index (CDI). The CDI is then a proxy of plant-soil feedbacks induced by taxonomic species during the length of the rotation [72,73]. Crop Diversity Index (CDI) is a modified version of Simpson's reciprocal diversity index [74] that reflects both the number of species and their relative abundance over the duration of the rotation, considering the temporal proportion of species. Briefly, this indicator multiplied the number of species in the rotation by the mean number of species grown per year over a five-year period (i.e., the longest rotation). Based on the crop diversity index, only two levels of diversity were determined LOW for values <3, and HIGH for values >3 for statistical powerness. Two different soil management was performed by farmers: a conventional tillage (CT) which refered to plowing depth of 25 cm or a reduced tillage (RT) when no plowing was performed (Table 1).

$$CDI = \frac{1}{\sum_{i=1}^{c} Pi^2}$$

where C is the number of species, and Pi is the proportion of the duration of the crop rotation that the $i^{th}$ species is present.

### 2.2. Soil sampling

The rhizosphere effect was calculated as the enzyme activity of rhizosphere soil minus that of standardized bulk soil.

To this end, enzyme activities were determined for both soil rhizosphere and bulk soils. Rhizosphere soils were sampled from study sites at the heading stage of wheat development [75] by collecting soil that adhered to the roots in a randomly placed 3 m × 3 m sampling zone. In each zone, four soil blocks (25 cm × 25 cm × 12 cm) were removed from four randomly placed locations to collect rhizosphere samples (i.e., four replicates). After excavation in the field, the soil blocks with their wheat plants were placed in a hand-held cooler (4°C) for transport to the laboratory. In the laboratory, plants were shaken vigorously to remove loose soil. Rhizosphere soils were carefully collected by selecting small aggregates

**Table 1. Description of the farm fields studied in Brittany, France. The wheat (Triticum aestivum) during the experiment and its place during the rotation is indicated in bold.**

| Site | S1 | S2 | S3 | S4 | S5 | S6 |
|---|---|---|---|---|---|---|
| Agricultural practices | | | | | | |
| Soil tillage | Conventional | Reduced | Reduced | Conventional | Conventional | Reduced |
| Rotation | wheat – IRG+ - maize – **wheat** | wheat – IRG+ - wheat – IRG+ - maize – **wheat** | Cover crop-maize – **wheat** | pea – wheat – barley – cover crop – maize – **wheat** | barley – rapeseed – wheat – cover crop – pea – cover crop – maize – **wheat** | rapeseed – wheat – HRG+/ red clover - pea/wheat/ vetch - maize – **wheat** |
| Cover crop | – | – | phacelia/ clover | oat/phacelia | oat/phacelia | – |
| Crop diversity index | 1.50 | 1.50 | 2.66 | 6.00 | 8.16 | 8.16 |
| Crop diversification level | Low | Low | Low | High | High | High |
| Wheat management | | | | | | |
| N fertilization (kg ha$^{-1}$) | 140 | 107 | 139 | 157 | 140 | 146 |
| P fertilization (kg ha$^{-1}$) | 0 | 30 | 55 | 19 | 0 | 55 |
| K fertilization (kg ha$^{-1}$) | 0 | 60 | 0 | 0 | 0 | 0 |
| Yield (t ha$^{-1}$) | 8.8 | 7.4 | 8.0 | 7.6 | 8.3 | 7.8 |
| Soil properties | | | | | | |
| Texture | Loamy clay | Loamy clay | Loamy clay | Loamy clay | Loamy clay | Loamy clay |
| Bulk density | 1.29 | 1.28 | 1.36 | 1.35 | 1.13 | 1.34 |
| pH$_{H2O}$ | 6.47 | 6.77 | 6.16 | 6.62 | 6.47 | 6.97 |
| P$_{Olsen}$ (g kg$^{-1}$) | 0.069 | 0.067 | 0.099 | 0.080 | 0.075 | 0.160 |

+Note: IRG = Italian ryegrass; HRG = hybrid ryegrass

(< 1 cm, corresponding to the maximum root influence based on the nutrient gradient [76] that still adhered to wheat roots after shaking. Enzyme activities of a composite rhizosphere of the plants in each block were analyzed within two days to capture the influence of the plants on microbial activity. Maximum basal enzyme activities without the influence of plant were determined and considered as bulk soil baselines to limit confounding effects of temperature and water content [77–79] on enzyme activities. Bulk soil baselines of each soil were determined after a short incubation of a bulk soil composite, based on recommendations for determining basal respiration in Europe [80]. The bulk soil composite was obtained after the wheat harvest in August (summer). We used an auger (15 cm in depth, 5 cm in diameter) at 20 points to collect soil samples at a depth 2–12 cm in the sampling zone, near the soil blocks that had been used to collect rhizosphere soils. Bulks soils were air-dried before determining soil enzyme activities. For both rhizosphere and bulk baseline soils, the remaining roots were hand-sorted in containers before determining enzyme activities.

## 2.3. Bulk soil baselines

Since the objective of this study was to determine a relative rhizosphere effect, a baseline of maximal enzyme activities in absence of plant influence was determined. The baseline is the reference state of soil enzyme in soil in optimal conditions (temperature, water) in absence of plants. This determination was performed from collecting soil composite on each field after harvest in July 2021. Four replicates of 150 g were taken from the composite samples collected after harvest. To reduce aggregate breakdown, replicates were then gently remoistened using a spray to 80% of the water holding capacity previously determined by the pressure-plate method [81]. Soils were then homogeneously mixed and incubated for a

short period (6 h) to prevent microbial growth that could have increased soil enzyme activity [82], and later incubated for one week at 15°C in airtight pots that contained a water-filled container to keep water from evaporating from the soil. After 7 days of incubation, 50 g of soil were collected and analyzed to determine enzyme activities.

## 2.4. Enzyme activity analyses

After collection of soils (either rhizosphere or bulk baseline) enzyme activities were all measured within 48 h at the Biochem-Env platform (INRAE, Versailles, France, http://www.biochemenv.fr/). For each soil sample (rhizosphere or bulk soil), the analyses were performed in three technical replicates by weighing 4 g of sieved soil [83] and mixing each with 25 mL of water for β-glucosidase, 25 mL of 50 mM Tris base pH 7.5 for arylamidase or 25 mL of 50 mM Tris HCl pH 5.5 for acid phosphatase. Soil suspensions were pipetted into 96-well microplates and maintained at 37°C. A specific substrate was added to the suspension depending on the enzyme tested: 0.05 4-nitrophenyl β-D-glucopyranoside for 1 h for β-glucosidase, 0.008 M of L-leucine b-naphtylamide for 2 h for arylamidase and 0.05 M 4-nitrophenylphosphate for 30 min for acid phosphatase. For β-glucosidase and acid phosphatase, the reaction was stopped with 0.5 M CaCl2 and 0.1 M Tris at pH 12. For arylamidase, the reaction was stopped with pure ethanol, and the b-Naphtylamine formed was colored with 0.0035 M of DMCA (p-dimethylaminocinnamaldehyde). The microplates were centrifuged after reactions for 5 min at 2000 g, and absorbance of the reaction product was measured on a spectrophotometer (SAFAS, Xenius, Monaco). The amounts of p-nitrophenol and b-Naphtylamine formed were estimated by absorbance at 405 nm and 540 nm, respectively. Enzyme activities equaled the amount of product formed per unit of time and were expressed in mU g$^{-1}$ dry soil (i.e., nmol of product released per minute per g of dry soil).

## 2.5. Rhizosphere Effect Indicator: theory and calculation

Because of the complexity of soil-plant interaction, it is often difficult to clearly identify the factors responsible for the variations in enzymatic activities and statistical analysis may suffer from confounding effects. In our study, variations in enzymatic activities can be caused by the rhizosphere effect, but also by variations in crop diversification, soil tillage and by global difference between study sites. To estimate the difference in functioning between the rhizosphere and bulk soils, we used linear mixed models (LMM) that allow to decompose each effect and test the rhizosphere effect without confusion. LMM also allow to investigate the rhizosphere effect at various level going from an overall rhizosphere effect to a effect within crop diversification levels and soil tillage.

**2.5.1. Theory.** The variability in the activity of each enzyme (Equation 1) was estimated for each enzymatic activity (Y) (arylamidase, β-glucosidase and acid phosphatase):

$$Y_{ijklm} = \mu + \alpha_i + \beta_j + \gamma_k + \alpha\beta_{ij} + \alpha\gamma_{ik} + \beta\gamma_{jk} + \alpha\beta\gamma_{ijk} + D_l + \varepsilon_{ijklm} \tag{1}$$

where μ is the intercept of mean enzyme activity; $\alpha_i$ is the marginal effect of the soil compartment (i = rhizosphere soil (RS) or bulk soil (BS)); $\beta_j$ is the marginal effect of crop diversification (j = High or Low); $\gamma_k$ is the marginal effect of soil tillage (k = reduced tillage (RT) or conventional tillage (CT)); $\alpha\beta_{ij}$, $\alpha\gamma_{ik}$, $\beta\gamma_{jk}$ and $\alpha\beta\gamma_{ijk}$ are marginal effects of two-way and three-way interactions between the soil compartment, diversification level and soil tillage; and $D_l$ is the marginal effect of the study site.

The various coefficients introduced in the model formalized in Equation 1 allow to account for all the potential effects that influence the variability of enzymatic activities, therefore preventing from confounding effects.

**2.5.2. Calculation.** Model parameters were estimated using the restricted maximum likelihood algorithm implemented in the *lme4* R package [84]. The estimated coefficients were first used to calculate the amount of an overall "rhizosphere effect indicator" (REI), which equaled to the difference between the enzyme activity in the rhizosphere and in the bulk soil:

$$REI = \alpha_{RS} - \alpha_{BS} \tag{2}$$

REI is then defined as the statistical contrasts between rhizosphere and bulk soil.

To study the rhizosphere effect within each crop diversification, we used a similar scheme and defined the following REIs ($REI_{High}$ and $REI_{Low}$ within *High* and *Low* diversification level):

$$REI_{Low} = \alpha_{RS}\beta_{Low} - \alpha_{BS}\beta_{Low} \tag{3}$$

$$REI_{High} = \alpha_{RS}\beta_{High} - \alpha_{BS}\beta_{High} \tag{4}$$

Similarly, we defined a REI for each soil tillage, namely $REI_{RT}$ and $REI_{CT}$ for calculating the Rhizosphere effect given reduced tillage and conventional tillage respectively:

$$REI_{RT} = \alpha_{RS}\gamma_{RT} - \alpha_{BS}\gamma_{RT} \tag{5}$$

$$REI_{CT} = \alpha_{RS}\gamma_{CT} - \alpha_{BS}\gamma_{CT} \tag{6}$$

Finally, using three-way interaction coefficients, REI were calculating for each combination of diversification level and soil tillage as follows:

$$REI_{Low,RT} = \alpha_{RS}\beta_L\gamma_{RT} - \alpha_{BS}\beta_L\gamma_{RT} \tag{7}$$

$$REI_{Low,CT} = \alpha_{RS}\beta_L\gamma_{CT} - \alpha_{BS}\beta_L\gamma_{CT} \tag{8}$$

$$REI_{High,RT} = \alpha_{RS}\beta_H\gamma_{RT} - \alpha_{BS}\beta_H\gamma_{RT} \tag{9}$$

$$REI_{High,CT} = \alpha_{RS}\beta_H\gamma_{CT} - \alpha_{BS}\beta_H\gamma_{CT} \tag{10}$$

**2.5.3. Significance of rhizosphere effect.** We then used the *lmerTest* package [85] to test for the significance of marginal and interaction effects in the model. Based on the LMMs, all REIs were calculated as the least-square mean of enzyme activity in the rhizosphere minus that in the bulk soil using the *emmeans* package [86]. The significance of each rhizosphere effect was compared to 0, which corresponded to the null hypothesis of the absence of a rhizosphere effect. Rhizosphere effects calculated as a function of crop diversification and type of soil tillage were further compared using the *emmeans* package. According to this method, a significant rhizosphere effect ($P < 0.05$) with a positive value indicated that the rhizosphere had higher enzyme activity than the bulk soil. Finally, we assessed the difference in REI between two related experimental conditions (i.e., high and low crop diversification, conventional and reduced tillage, LowRT and HighRT, and LowCT and HightCT) using pairwise comparisons (Tukey's range test) as implemented in the *emmeans* package.

**2.5.4. Calculation of an integrated rhizosphere effect as a proxy for actionable knowledge.** We calculated an integrated REI that combined all three enzyme activities measured to provide an overview of the rhizosphere effect and facilitate interpretation of soil processes for decision-making [73]. To this end, a first step was to perform a principal component analysis (PCA) on the three enzyme activities (*FactoMineR* package [87]). This step is mandatory to define the integrated REI as the coordinate of each soil sample on the first component of the PCA. We then used in a second step the same LMM procedure to calculate and test rhizosphere effects. All statistical analyses were performed using R software [88], version 4.0.3.

## 3. Results

### 3.1. Intensity of the rhizosphere effect per soil enzyme in the context of crop-livestock farming

Mean enzyme activities were higher in the rhizosphere than in the bulk soil for all treatments (Fig 1, Table 2). Activity of arylamidase (Fig 1a) was lower (< 11 mU g$^{-1}$ of dry soil) than those of β-glucosidase and acid phosphatase (< 40 mU and < 60 mU g$^{-1}$ dry soil, respectively) (Fig 1b,c). Activity was highest for β-glucosidase in the HighRT treatment (34.23 mU g$^{-1}$ of dry soil), which was a mean of 3.5 times higher than those in the other treatments. Arylamidase and β-glucosidase in both the rhizosphere and bulk soil increased when the diversification level shifted from low to high, while other treatments exhibited different patterns (Fig 1a,b). The integrated soil variable was lowest (and negative) in the HighCT treatment for a bulk soil and highest (and positive) in the HighRT treatment for a rhizosphere soil (Fig 2). The integrated soil variable was lowest in the LowCT treatment and highest in the HighRT treatment. The overall rhizosphere effect was positive and differed significantly from 0 for all enzymes (Table 2). The models estimated an REI of 6.59 for β-glucosidase, 3.03 for arylamidase and 11.3 for acid phosphatase when only the observations by soil compartment were compared. When analyzing only the high diversification treatment, the rhizosphere effect was significant for all enzyme activities: 12.53 for the β-glucosidase, 3.93 for arylamidase and 18.48 for acid phosphatase (Table 2). In the low diversification treatment, however, the rhizosphere effect was significant for arylamidase (2.13) and acid phosphatase activities (4.19) but not for β-glucosidase activity (0.64) (Table 2), which differed little between the rhizosphere and bulk soils (Fig 1). Regarding the influence of soil tillage, the rhizosphere effect was significant for both RT and CT for all enzyme activities: respectively 11.51 and 1.67 for β-glucosidase, 4.45 and 1.61 for arylamidase and 14.13 and 8.54 for acid phosphatase (Table 2). When analyzing the combination of both practices (crop diversification × soil tillage), the rhizosphere effect was significant for all enzyme activities for HighRT and HighCT (Table 2). For arylamidase, the rhizosphere effect was also significant for LowRT (3.31) and LowCT (0.95). For β-glucosidase, however, the rhizosphere effect was not significant for LowRT (1.39) or LowCT (−0.11). For acid phosphatase, the rhizosphere effect was significant for LowRT (4.48) but not for LowCT (0.32).

### 3.2. Combined effects on crop diversification and soil tillage on the rhizosphere effect

Comparing estimates of the rhizosphere effect between each treatment tested separately, the rhizosphere effect was significantly higher for high diversification than for low diversification and for reduced tillage than for conventional tillage for all three enzymes (Table 3).

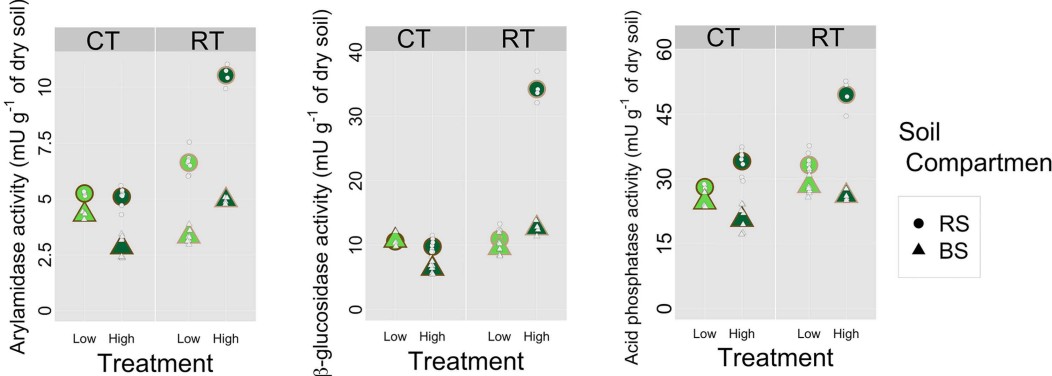

**Fig 1. Measured (lower symbols) and means (upper symbols) of (a) arylamidase, (b) β-glucosidase and (c) acid phosphatase activities in (d) bulk soil (BS) and rhizosphere (RS) for each crop diversification level (Low and High) and each type of soil tillage (conventional (CT) and reduced (RT)).**

**Table 2. Model estimates and P values of the overall rhizosphere effect and the rhizosphere effect given each level of crop diversification (High or Low), soil tillage (conventional (CT) or reduced (RT)) or each combination of both obtained by pairwise comparisons between means estimated by the models for arylamidase, β-glucosidase and acid phosphatase and the integrated rhizosphere effect indicator.**

| Experimental condition | Arylamidase | | β-glucosidase | | Acid phosphatase | | Integrated | |
|---|---|---|---|---|---|---|---|---|
| | Estimate | P value | Estimate | P value | Estimate | P value | Estimate | P value |
| Overall | 3.03 | 0.000* | 6.59 | 0.000* | 11.30 | 0.000* | **2.21** | ***0.000*** |
| Crop diversification | | | | | | | | |
| High | 3.93 | <0.001* | 12.53 | <0.001* | 18.48 | <0.001* | **3.46** | ***<0.001*** |
| Low | 2.13 | <0.001* | 0.64 | 0.468 | 4.19 | <0.001* | **0.95** | ***<0.001*** |
| Tillage | | | | | | | | |
| Reduced (RT) | 4.45 | <0.001* | 11.51 | <0.001* | 14.13 | <0.001* | **3.21** | ***<0.001*** |
| Conventional (CT) | 1.61 | <0.001* | 1.67 | 0.003* | 8.54 | 0.000* | **1.21** | ***<0.001*** |
| Diversification × Tillage | | | | | | | | |
| HighRT | 5.59 | <0.001* | 21.62 | <0.001* | 23.42 | <0.001* | **5.02** | ***<0.001*** |
| HighCT | 2.27 | <0.001* | 3.44 | <0.001* | 13.54 | <0.001* | **1.90** | ***<0.001*** |
| LowRT | 3.31 | <0.001* | 1.39 | 0.136 | 4.84 | 0.002* | **1.39** | ***<0.001*** |
| LowCT | 0.95 | 0.021* | −0.11 | 1.000 | 3.54 | 0.321 | **0.51** | ***0.140*** |

\* identifies significant differences with associated P values, while grey text indicates non-significant results.

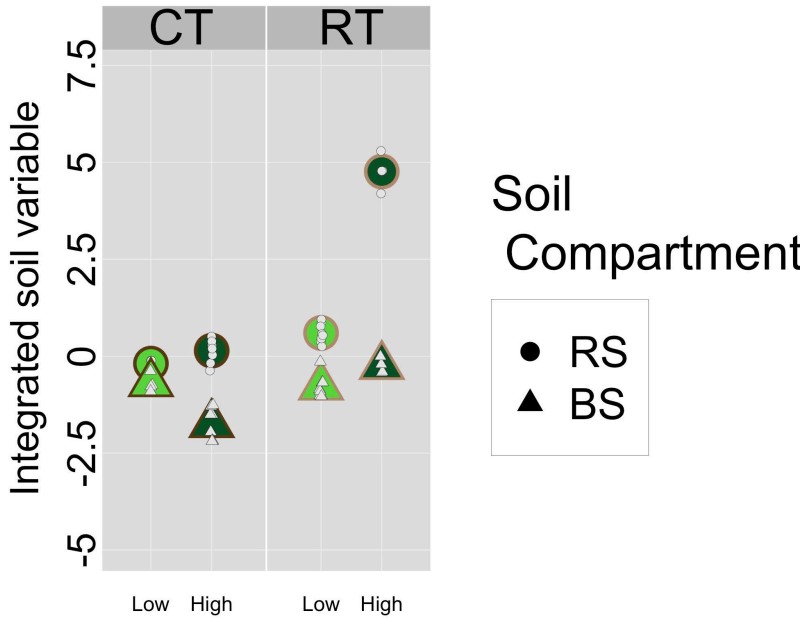

**Fig 2. Information of the soil variables integrated into from the first component of the PCA analysis.**

The difference between the rhizosphere effect of high and low diversification was 11.9, 1.80 and 14.29 for β-glucosidase, arylamidase and acid phosphatase, respectively (Table 3). Compared to conventional tillage, reduced tillage increased the rhizosphere effect by 9.84, 2.84 and 5.59 for β-glucosidase, arylamidase and acid phosphatase,

**Table 3. Model estimates and P values of pairwise comparisons of the rhizosphere effect between each level of crop diversification (High or Low), soil tillage (conventional (CT) or reduced (RT)) or each combination of both for arylamidase, β-glucosidase and acid phosphatase and the integrated rhizosphere effect indicator.**

| Experimental condition | Arylamidase | | β-glucosidase | | Acid phosphatase | | Integrated | |
|---|---|---|---|---|---|---|---|---|
| | Estimate | P value | Estimate | P value | Estimate | P value | estimate | P value |
| Crop diversification | | | | | | | | |
| High – Low | 1.80 | <0.001* | 11.89 | <0.001* | 14.29 | <0.001* | **2.51** | **<0.001*** |
| Compartment × Tillage | | | | | | | | |
| RT – CT | 2.84 | <0.001* | 9.84 | <0.001* | 5.59 | 0.002* | **2.00** | **<0.001*** |
| Crop diversification × Tillage | | | | | | | | |
| HighRT -LowRT | 2.29 | <0.001* | 20.23 | <0.001* | 18.58 | <0.001* | **3.64** | **<0.001*** |
| HighRT – HighCT | 3.33 | <0.001* | 18.18 | <0.001* | 9.88 | 0.002* | **3.12** | **<0.001*** |
| HighRT – LowCT | 4.65 | <0.001* | 21.74 | <0.001* | 19.89 | <0.001* | **4.51** | **<0.001*** |
| LowRT – HighCT | 1.04 | 0.064 | −2.05 | 0.472 | −8.70 | 0.001* | **−0.51** | **0.560** |
| LowRT – LowCT | 2.36 | <0.001* | 1.50 | 0.991 | 1.31 | 1.000 | **0.88** | **0.075** |
| HighCT – LowCT | 1.32 | 0.045* | 3.56 | 0.041* | 10.01 | 0.001* | **1.39** | **<0.001*** |

* identifies significant differences with associated P values, while grey text indicates non-significant results

respectively (Table 3). Comparing estimates of the rhizosphere effect for the combinations of crop diversification × soil tillage, HighRT had a significantly higher rhizosphere effect than all other combinations, for all three enzymes (Table 3). The largest difference in the rhizosphere effect was between HighRT and LowCT, with estimated values of 21.74, 4.65 and 19.89 for β-glucosidase, arylamidase and acid phosphatase, respectively (Table 3).

In detail, beginning with the overall rhizosphere effect, the rhizosphere effect increased as crop diversification increased for all three enzymes (Fig 3). In fields with high crop diversification, the rhizosphere effect was significantly higher for reduced tillage than conventional tillage for all three enzymes (Fig 3). In fields with low crop diversification, however, the rhizosphere effect was significantly higher for reduced tillage than conventional tillage only for arylamidase (Fig 3). Moreover, shifting from conventional to reduced tillage on fields with low crop diversification did not significantly increase the rhizosphere effect for β-glucosidase or acid phosphatase (Fig 3). Interestingly, for arylamidase activity, the rhizosphere effect increased significantly with low diversification when reduced tillage was implemented. Finally, with high diversification, the rhizosphere effect decreased significantly when shifting from reduced to conventional tillage, especially for β-glucosidase (Fig 3), while for arylamidase and acid phosphatase, it reached a mean similar to that of the overall rhizosphere effect.

### 3.3. A proxy of the rhizosphere effect for actionable knowledge

The integrated REI summarized the individual enzyme activities, with significant rhizosphere effects estimated by the LMMs, except for the LowCT treatment, which was consistent with the individual enzyme activities estimated. The rhizosphere effect of the integrated REI was significantly higher for HighRT than for the other three combinations of practices (Table 3, Fig 2). The largest differences were for LowCT, while LowRT and HighCT had intermediate rhizosphere effects that did not differ significantly (Fig 2). Under conventional tillage, REI was significantly higher for high diversification than for low diversification; however, no significant differences were estimated between the rhizosphere effects of HighCT and the LowRT (Table 3).

## 4. Discussion

### 4.1. Theory and calculation of the rhizosphere effect indicator

We developed a field REI based on mean differences in soil enzyme activities in two soil compartments: the rhizosphere and bulk soil not influenced by root activities. Its main characteristics are (i) statistical determination of paired

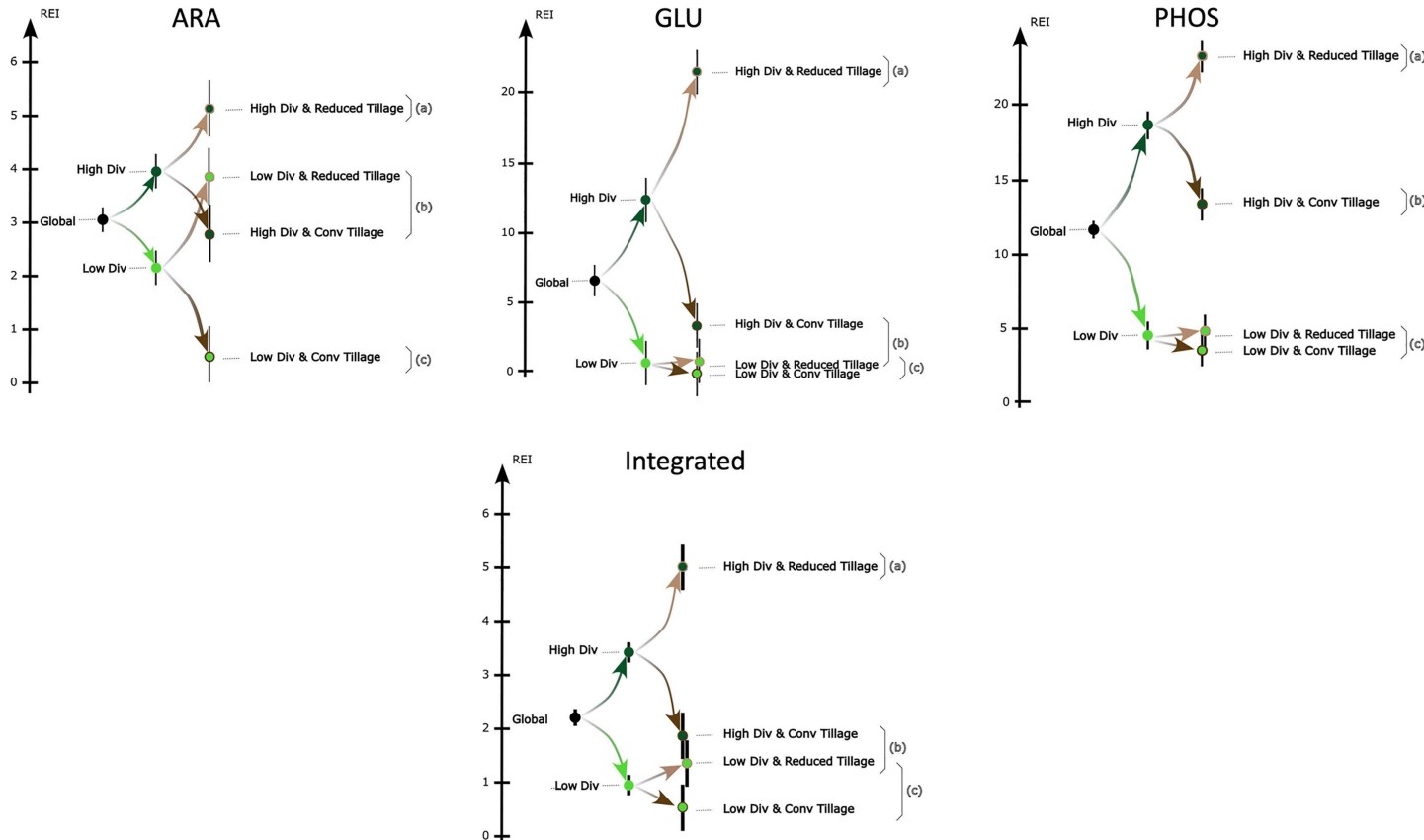

**Fig 3. Arylamidase (ARA), β-glucosidase (GLU), acid phosphatase (PHOS) and the integrated rhizosphere effect indicator (Integrated) for each combination of crop diversification level (Low or High) and soil tillage (conventional (CT) or reduced (RT)).** Different letters indicate significantly different pairwise comparisons between treatments.

samples, (ii) detailed analysis of the rhizosphere effect in order to assess the influence of agricultural practices and (iii) determination of baselines considered as maximum potential enzyme activities. The REI is based only on soil enzymes considered to be sensitive to simple and inexpensive methods [60] and then adapted for use and interpretation by laypeople. The REI costs less to determine than other *in situ* methods based on molecular analysis [60], previous knowledge to determine nutrient flows in the rhizosphere [89] or determination of enzyme activities in fields [90]. The procedure used to determine enzyme activities of bulk soil could be questioned, but we considered several potential advantages and disadvantages, especially the storage conditions, rewetting before incubation and the duration and temperature of the incubations. Since many factors, such as land use, climate, soil properties and the type of enzyme, can influence the resilience of enzymes to storage conditions, the best practice is to clearly define the storage conditions, as stated by [91] in the absence of standardized methods. Rewetting of bulk soil was chosen because it was considered to have the same disadvantages as freezing on mineralization rates [92]. The pre-incubation duration was set as a trade-off between soil respiration and microbial growth of dormant cells [93]. The pre-incubation temperature and duration were based on comparing basal respiration across Europe [80]. Since the objective of this study was to determine differences among treatments rather than absolute enzyme activities, we were able to statistically determine rhizosphere effects and evaluate effects of agricultural practices on the rhizosphere effect in order to promote peer discussion.

## 4.2. Effects of agricultural practices on the rhizosphere effect

The results shed light on the contrasting results for stimulation of the rhizosphere effect by agricultural practices. The maximum rhizosphere effect for all enzymes was estimated for high crop diversification and reduced tillage. However, high crop diversification did not maintain this maximum rhizosphere effect on arylamidase when soil tillage increased, unlike those on β-glucosidase and acid phosphatase. The observed negative effect of conventional tillage of bulk soil on arylamidase activity was also noted in other studies [94]. To begin, a strong negative correlation has been observed between total N content and arylamidase activity [95]. We thus hypothesized that plowing re-homogenizes the distribution of soil organic N in the plowed layer, which maintains high N availability throughout the topsoil, while reduced tillage concentrates soil organic matter in the few first cm of soil. In turn, arylamidase production by soil microbes in the plowed layer could have been decreased, thus decreasing the benefit of diversification for the arylamidase rhizosphere effect. Results of the present study did not support those of other studies that indicated higher β-glucosidase activity with high crop diversification in bulk soil [96]. Given the weak but significant arylamidase rhizosphere effect, we thus hypothesized that plant with their microbial consortia in root vicinites mine N from soil organic matter. In accordance with local fertilization standards, the sites had low (site S6) to moderately high (site S3) available P contents in the soil. Consequently, the rhizosphere effect observed in this study was attributed to fertilizer management that maintained relatively low P availability and promoted the rhizosphere effect by inducing production of acid phosphatase. However, further experiments are necessary to fully understand this study's results for the rhizosphere effect, but our results converges towards the possible development of a bulk soil microbiome by crop diversification [97] determining the rhizosphere effect but is easily weaken when tillage shifted from reduced to conventional tillage.

## 4.3. Potential application and perspectives

Under climate change, projections estimate an increase in phosphate genes in the bulk soil microbiome [98], suggesting that the potential of the rhizosphere effect should continue to be explored, especially for the critical finite resource of P [99]. However, we highlighted that the legacy effect of N, P fertilization should also be considered when one is considering the promise of rhizosphere effect in future breeding programs [97]. Soil inoculation of seeds or plants [100] and nanomaterial fertilizers [101] are also recommended to sustain agricultural productivity. With this broader perspective, we however plead to consider the basics, which means that the length of crop rotation, soil management and fertilization as a set of agricultural practices for managing the rhizosphere effect. In the same vein, emerging discourses recommend developing "slow tools" [102] that allow farmers to work the land carefully but effectively to induce transition towards agroecological practices. The integrated REI developed in the present study could contribute to this goal, but further developments are required to explicitly describe the influence of the rhizosphere effect on plant performances [103]. This consideration envision the cognitive consideration of long-term agricultural productivity and legacy effect in agriculture nature-based solutions to consider.

Stakeholders usually play unsignificant role in indicator development that are designed to improve the focal system as soil [60]. However, agriculture nature-based actions are efficient if the understanding and engagement is shared among stakeholders to avoid sincere but misleading decisions. Our farmer's panel was receptive to the rhizosphere knowledge but in turn plead for giving them information rather than a new tool.

## 5. Conclusion

Our approach is a thinking-transitioning approach towards more radical agroecologization. We demonstrated that the rhizosphere effect, a component of plant-soil feedback, can be effectively managed. Fertilization and soil management practices can be adapted immediately without waiting new cultivars or shifting with non-mature technologies

## Acknowledgments

The authors warmly thank the farmers in northwestern France – Guillaume Gasc and Bertrand Pinel – for their help in contacting other farmers. This study was possible due to the active participation of the technical staff of UMR SAS, especially Claire Goetz, who collected the dataset.

## Author contributions

**Conceptualization:** Edith Le Cadre, Mathieu Emily.

**Data curation:** Sebastian Mira.

**Formal analysis:** Mathieu Emily.

**Funding acquisition:** Edith Le Cadre.

**Methodology:** Edith Le Cadre, Mathieu Emily.

**Supervision:** Edith Le Cadre, Mathieu Emily.

**Validation:** Mathieu Emily.

**Writing – original draft:** Edith Le Cadre.

**Writing – review & editing:** Edith Le Cadre, Xiaoyan Tang, Mathieu Emily.

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
