## [Decision Letter · Decision Letter 0]

10 Apr 2025

PONE-D-24-30422
Choreographing rhizosphere effect with agricultural practices for agroecology ?
PLOS ONE

Dear Dr. Le Cadre,

Thank you for submitting your manuscript to PLOS ONE. After careful consideration, we feel that it has merit but does not fully meet PLOS ONE’s publication criteria as it currently stands. Therefore, we invite you to submit a revised version of the manuscript that addresses the points raised during the review process.

Dear Author
As you can see there is 2 reviewers that ask for minor revisions, however, one of the reviewers rejected your paper.
Please address all reviewers comments with a special focus on reviewer 2 comments.
Good luck and waiting for your feedback
Regards

We look forward to receiving your revised manuscript.

Kind regards,

Rachid Bouharroud

Academic Editor

PLOS ONE

Journal Requirements:

4. In the online submission form, you indicated that your data will be submitted to a repository upon acceptance.  We strongly recommend all authors deposit their data before acceptance, as the process can be lengthy and hold up publication timelines. Please note that, though access restrictions are acceptable now, your entire minimal  dataset will need to be made freely accessible if your manuscript is accepted for publication. This policy applies to all data except where public deposition would breach compliance with the protocol approved by your research ethics board. If you are unable to adhere to our open data policy, please kindly revise your statement to explain your reasoning and we will seek the editor's input on an exemption.

Reviewers' comments:

Reviewer's Responses to Questions

**Comments to the Author**

1. Is the manuscript technically sound, and do the data support the conclusions?

Reviewer #1: Yes

Reviewer #2: No

Reviewer #3: Yes

2. Has the statistical analysis been performed appropriately and rigorously? 

Reviewer #1: Yes

Reviewer #2: I Don't Know

Reviewer #3: Yes

3. Have the authors made all data underlying the findings in their manuscript fully available?

Reviewer #1: Yes

Reviewer #2: Yes

Reviewer #3: Yes

4. Is the manuscript presented in an intelligible fashion and written in standard English?

Reviewer #1: Yes

Reviewer #2: No

Reviewer #3: Yes

5. Review Comments to the Author

Reviewer #1: 1.Please explain why four replicates, rather than three, are used for sampling and research on enzyme activity.

2. The reference number in the text and the reference section must match. Line 213 for Mira et al., 2022, however, has some ambiguity.

3. Substitute the appropriate reference number.

Reviewer #2: My main concern with this manuscript is the samples used. There is a comparison made between rhizosphere soil collected from wheat roots at the "heading stage", and bulk soil collected at an entirely different time - at or after harvest. Furthermore the two soil types are treated quite differently. Enzymes are measured on the rhizosphere soil within 2 days, on the bulk soilafter wetting and incubaction. How can these two soil types be compared and how can any differences be attributed to a rhizosphere effect?

Furthermore the nature of the crops in the fields is unclear - in Table 1 the headers indicate the species used in the rotations -but the treatment do not seem comparable with respect to species compositions, and some things are not defined (i.e. what is IRG?) - how then can differences between wheat root rhizospheres be attributed to the other cropping procedures if the species mixes are NOT the same?

Finally, the complex REI is derived by fitting the data to statistical model that assumes there is a tillage and a diversity effect. I find this difficult to understand - why don't the authors just present the actual enzymatic activity and show statistically significant difference.

I believe the first two issues are important and negate any explanation of why the REI model was used. In addition, the english needs some improvement - particularly around the use of plurals and articles. Some basic definitions are lacking - i.e. of "tempora diversification" in the abstract - and of "crop diversification process on line 66. There is no clear description of the species growing in these fields.

Reviewer #3: The paper addresses a key issue in agroecology: the rhizosphere effect and its interaction with agricultural practices.

The methodology combines enzymatic analyses (β-glucosidase, arylamidase, acid phosphatase) with agronomic parameters. The use of indicators such as the Rhizosphere Effect Index (REI) provides valuable insights that could be applicable in assessing soil health and optimizing sustainable practices.

Specific comments: Redesign figures with larger fonts, distinct colors, and clearer legends.

6. PLOS authors have the option to publish the peer review history of their article (what does this mean?). If published, this will include your full peer review and any attached files.

Reviewer #1: No

Reviewer #2: No

Reviewer #3: No

---

## [Author Response · Author response to Decision Letter 1]

5 Jun 2025

Please download the formatted version uploaded for a better reading experience

Manuscript PONE-D-24-30422 – responses to reviewer comments

Authors responses in blue.

Comments to the Author

1. Is the manuscript technically sound, and do the data support the conclusions?

Reviewer #1: Yes

Reviewer #2: No

We hope that our answers and modifications fully address Reviewer’s #2 concerns about the innovative approach that combined enzyme activities into an indicator to study the rhizosphere effect in the field, as a way to scale out application of studies of micro- and mesocosms extensively used to describe and understand the rhizosphere effect.

Reviewer #3: Yes

2. Has the statistical analysis been performed appropriately and rigorously?

Reviewer #1: Yes

Reviewer #2: I Don’t Know

We are happy to address Reviewer #2’s concerns about the statistical analysis, which was performed under the supervision of Dr. Mathieu Emily, a full professor in statistics (https://emily.perso.math.cnrs.fr/Homepage/index.htm). We have addressed the comments in the present manuscript point by point and modified the manuscript to increase its clarity.

Reviewer #3: Yes

3. Have the authors made all data underlying the findings in their manuscript fully available?

Reviewer #1: Yes

Reviewer #2: Yes

Reviewer #3: Yes

4. Is the manuscript presented in an intelligible fashion and written in standard English?

Reviewer #1: Yes

Reviewer #2: No

Since Reviewer #2 did not identify any specific English errors to guide our revision, we reviewed the English of the entire manuscript. In addition, a native English speaker, who is also a senior scientist, verified the revised manuscript’s English (proofreading certificate submitted online). Accordingly, we are confident that Reviewer #2 will be satisfied by the English of the revised manuscript.

Reviewer #3: Yes

5. Review Comments to the Author

REVIEWER 1

Reviewer #1: 1.Please explain why four replicates, rather than three, are used for sampling and research on enzyme activity.

The number of replicates is a key factor that determines the statistical power of an experiment. While three replicates are the absolute minimum for statistical analysis, the optimal number is a trade-off between feasibility and the ideal number of replicates needed to capture the variability and size effect. In the study’s experiment, four replicates was the maximum possible and better than the bare minimum of three replicates. We decided not to address this detail in the revised manuscript, since we as scientists always have this trade-off in mind.

2. The reference number in the text and the reference section must match. Line 213 for Mira et al., 2022, however, has some ambiguity.

Yes, we agree; we modified the reference number accordingly.

3. Substitute the appropriate reference number.

Done

REVIEWER 2

Reviewer #2: My main concern with this manuscript is the samples used. There is a comparison made between rhizosphere soil collected from wheat roots at the "heading stage", and bulk soil collected at an entirely different time - at or after harvest.

Using a baseline to determine bulk enzymes was discussed extensively (LL 311-323) in the original manuscript. Nonetheless, we detail our method in this response and have modified the Materials and methods section.

Thank you for giving us the opportunity to describe our method in more detail. Our objective was to capture the mean rhizosphere effect to compare diversification patterns, but not to assess the real rhizosphere effect in the field, which varies daily if not hourly due to rhizodeposition patterns influenced by crop physiology. Regarding assessment of the true rhizosphere effect, we described the method we used in a previous article (Mira et al., 2022).

Mira S, Emily M, Mougel C, Ourry M, Le Cadre E (2022) A field indicator for rhizosphere effect monitoring in arable soils. Plant and Soil. doi:10.1007/s11104-021-05284-2

This article (Mira et al, 2022), which we cited in the present manuscript, extensively debated the advantages and disadvantages of the method suggested by reviewer #2. In brief, our method captures dynamics of the rhizosphere effect over time by collecting rhizosphere and bulk soils on the same day. However, we have addressed concerns in the present manuscript about differences in the dynamics of water content and temperature caused by differences in weather among sites, which have legacy effects on soil enzymes.

In addition to rhizodeposition, rainfall variability and soil temperature due to albedo (among others) differ between rhizosphere soil and bulk soil and influence enzyme activities differently. These factors induced legacy effects whose durations are impossible to predict.

For example, the physical modification caused by the reorganization of soil aggregates around roots does not influence bulk soil. In particular, mucilage around roots helps to maintain a hydraulic connection, which results in higher water content around roots than that in a drying bulk soil (Helliweell et al., 2017).

Helliwell JR, Sturrock CJ, Mairhofer S, Craigon J, Ashton RW, Miller AJ, Whalley WR, Mooney SJ (2017) The emergent rhizosphere: imaging the development of the porous architecture at the root-soil interface. Scientific Reports 7 (1):14875. doi:10.1038/s41598-017-14904-w

Moreover, the size of aggregates that one can collect around roots depends mainly on the water and clay contents which may slightly differ among sites even at similar clay content captured by soil analyses. Operationally, these differences can cause the size of aggregates to differ. This is a critical parameter to consider, since any differences in the size of aggregates interferes with the rhizosphere effect.

Accordingly, we decided to set a baseline of enzyme activity to assess how wheat roots can increase the inherent capacity of soils. Defining the baseline is important because it identifies the minimum enzyme activities that a soil can be expected to perform when environmental variables are optimal without plants. Accordingly, the baseline is the reference state of potential enzyme activities in the absence of plants. Soil incubation was set to provide optimal conditions for enzyme activity. We modified the Materials and methods section to highlight these key points better. In addition, as in our previous version, the pros and cons of the baseline is discussed section 4.1

Finally, the soil used to determine the baseline was not collected at or after harvest, as Reviewer’s #2 seems to have understood, but systematically after harvest (third week of July, this is indicated in the revised version).

LL 152 -155: Since the objective of this study was to determine a relative rhizosphere effect, a baseline of maximal enzyme activities in absence of plant influence was determined. The baseline is the reference state of soil enzyme in soil in optimal conditions (temperature, water) in absence of plants.

LL 156: This determination was performed from collecting soil composite on each field after harvest in July 2021.

 Furthermore the two soil types are treated quite differently.

We do not understand this comment. If Reviewer #2 is referring to soil types among sites (S1-S6), we attempted to carefully limit effects of soil types (whose properties could differ) by choosing plots with similar properties (Table 1). Moreover, the statistical model considered the remaining (marginal) site effect (Dl in Yijklm = µ + αi + βj + γk + αβij + αγik + βγjk + αβγijk + Dl + εijklm). This statistical modelling approach is state-of-art for comparing multiple sites when variability cannot be controlled, unlike the variability in factorial designs used in field or laboratory experiments.

Please note that table 2 was improved for sake of clarity

If Reviewer’s #2 is worried about how enzyme activities were determined, we address this specific comment below.

 Enzymes are measured on the rhizosphere soil within 2 days, on the bulk soil after wetting and incubaction. How can these two soil types be compared and how can any differences be attributed to a rhizosphere effect?

First, recall that our objective was not to analyze rhizosphere soil and bulk soil, as we did in our previous study (Mira et al., 2022), but to estimate the relative rhizosphere effect by calculating the difference between the two. This difference was compared among sites, and the statistical model distinguished effects of crop diversification (the crop diversity index (CDI)) and soil management on it.

All enzymes in rhizosphere soil and bulk soil were determined within 2 days of collection. We have modified the Material and methods section (LL 166) because this critical detail was lacking. We apologize and thank Reviewer #2 for pointing it out.

L 166: After collection of soils (either rhizosphere or bulk baseline) enzyme activities

 Furthermore the nature of the crops in the fields is unclear - in Table 1 the headers indicate the species used in the rotations -but the treatment do not seem comparable with respect to species compositions

We explain in the following responses why the treatments can be compared due to the experimental and statistical methods used, but also due to using the CDI, which was a proxy of the intensity of crop rotation (Low or High). We detail our arguments below.

 How then can differences between wheat root rhizospheres be attributed to the other cropping procedures if the species mixes are NOT the same?

Reviewer #2 is right that the rotations contained different species and different species mixtures over time for the same duration of rotation. This is why we used the CDI with two levels (Low and High), which reproduced a common approach to capture the diversity of plant species (Costa et al., 2024). The CDI is a modified version of Simpson's reciprocal diversity index (D) (Simpson, 1949) that reflects both the number of species and their relative abundance over the duration of a rotation, considering the temporal proportion of species:

CDI=1/(∑_(i=1)^c▒Pi²)

where C is the number of species, and Pi is the proportion of the duration of the crop rotation that the ith species is present.

The fact that the crop species cultivated before wheat was not the same was, however, considered by the CDI as a proxy of plant-soil feedback induced by individual crop species.

Costa A, Bommarco R, Smith ME, Bowles T, Gaudin ACM, Watson CA, Alarcón R, Berti A, Blecharczyk A, Calderon FJ, Culman S, Deen W, Drury CF, Garcia y Garcia A, García-Díaz A, Hernández Plaza E, Jonczyk K, Jäck O, Navarrete Martínez L, Montemurro F, Morari F, Onofri A, Osborne SL, Tenorio Pasamón JL, Sandström B, Santín-Montanyá I, Sawinska Z, Schmer MR, Stalenga J, Strock J, Tei F, Topp CFE, Ventrella D, Walker RL, Vico G (2024) Crop rotational diversity can mitigate climate-induced grain yield losses. Global Change Biology 30 (5):e17298. doi: https://doi.org/10.1111/gcb.17298

The levels “low” and “high” were determined after calculating the CDI (equation above). To maintain statistical power, we used only two levels for crop diversification, as we did for soil management.

LL 110-117: To capture the legacy effect of past crops modifying soil properties including the soil microbiome (70-73), we calculate the Crop Diversity Index (CDI). The CDI is then a proxy of plant-soil feedbacks induced by taxonomic species during the length of the rotation (70, 71). Crop Diversity Index (CDI) is a modified version of Simpson's reciprocal diversity index (72) that reflects both the number of species and their relative abundance over the duration of the rotation, considering the temporal proportion of species. Briefly, this indicator multiplied the number of species in the rotation by the mean number of species grown per year over a five-year period (i.e. the longest rotation). Based on the crop diversity index, two levels of diversity were determined LOW for values <3, and HIGH for values >3. Two different soil management was performed by farmers: a conventional tillage (CT) which refered to plowing depth of 25 cm or a reduced tillage (RT) when no plowing was performed (Table 1).

Finally, recall that only one crop species (wheat) was present when the soils were collected. Thus, we compared the rhizosphere effect induced only by wheat but that previous crops (assessed using the CDI) might also have influenced by modifying soil properties including the microbiome (D’Acunto et al., 2018; Sun et al., 2023; Kaloterakis et al., 2024, 2025).

Kaloterakis N, Giongo A, Braun-Kiewnick A, Rashtbari M, Zamberlan P, Razavi BS, Smalla K, Reichel R, Brüggemann N (2025) Rotational diversity shapes the bacterial and archaeal communities and confers positive plant-soil feedback in winter wheat rotations. Soil Biology and Biochemistry 203:109729. doi: https://doi.org/10.1016/j.soilbio.2025.109729

Kaloterakis N, Rashtbari M, Razavi BS, Braun-Kiewnick A, Giongo A, Smalla K, Kummer C, Kummer S, Reichel R, Brüggemann N (2024) Preceding crop legacy modulates the early growth of winter wheat by influencing root growth dynamics, rhizosphere processes, and microbial interactions. Soil Biology and Biochemistry 191:109343. doi:https://doi.org/10.1016/j.soilbio.2024.109343

Sun L, Wang S, Narsing Rao MP, Shi Y, Lian ZH, Jin PJ, Wang W, Li YM, Wang KK, Banerjee A, Cui XY, Wei D (2023) The shift of soil microbial community induced by cropping sequence affect soil properties and crop yield. Front Microbiol 14:1095688. doi:10.3389/fmicb.2023.1095688

D’Acunto L, Andrade JF, Poggio SL, Semmartin M (2018) Diversifying crop rotation increased metabolic soil diversity and activity of the microbial community. Agriculture, Ecosystems & Environment 257:159-164. doi:https://doi.org/10.1016/j.agee.2018.02.011

We understand that this criticism was due to a lack of clarity in the original manuscript. We hope that the modifications in the revised manuscript clarify Reviewer #2’s concerns.

 and some things are not defined (i.e. what is IRG?)

Table 1 of the original manuscript did have a footnote that described that IRG was Italian ryegrass and HRG was hybrid ryegrass. To increase clarity, we have added a symbol to Table 1 to indicate that IRG and HRG are defined in a footnote:

;Note: IRG = Italian ryegrass; HRG = hybrid ryegrass

 Finally, the complex REI is derived by fitting the data to statistical model that assumes there is a tillage and a diversity effect. I find this difficult to understand - why don’t the authors just present the actual enzymatic activity and show statistically significant difference.

Since we did not use the term “complex REI”, we think that Reviewer #2 is referring to the “integrated REI” (section 2.5.2). The integrated REI combines the information provided by the three enzymes to assess the overall rhizosphere effect. We believe that this is an innovative approach because the rhizosphere effect is considered an emergent property (Vetterlein et al., 2022) that cannot be deduced by analyzing individual variables.

Vetterlein D, Carminati A, Schnepf A (2022) Special issue: Rhizosphere spatiotemporal organisation: an integrated approach linking above and belowground. Plant and Soil 478 (1):1-4. doi:10.1007/s11104-022-05716-7

The method that Reviewer #2 suggests thus seems irrelevant for this study’s research question since it does not assess the overall rhizosphere effect as an emergent property of the soil-plant system. As described in a previous response, there are methodological and statistical constraints of comparing rhizosphere soil and bulk soil among multiple sites.

We used linear mixed models to break down the total variability in enzyme activities into multiple effects. In these models, we included and corrected for potential effects of crop diversification and soil tillage to ensure that the estimated rhizosphere effect was not confounded with other controlled factors. Moreover, including interaction effects in these models enabled us to estimate and test the rhizosphere effect at multiple levels: for

---

## [Editor Report · Decision Letter 1]

13 Jun 2025

Choreographing rhizosphere effect with agricultural practices for agroecology ?

PONE-D-24-30422R1

Dear Dr. Le Cadre,

We’re pleased to inform you that your manuscript has been judged scientifically suitable for publication and will be formally accepted for publication once it meets all outstanding technical requirements.

Kind regards,

Rachid Bouharroud

Academic Editor

PLOS ONE
---

## [Editor Report · Acceptance letter]

PONE-D-24-30422R1

PLOS ONE

Dear Dr. Le Cadre,

I'm pleased to inform you that your manuscript has been deemed suitable for publication in PLOS ONE. Congratulations! Your manuscript is now being handed over to our production team.

Kind regards,

on behalf of

Dr. Rachid Bouharroud

Academic Editor

PLOS ONE